# Review of the Pathogenic Mechanism of Grape Downy Mildew (*Plasmopara viticola*) and Strategies for Its Control

**DOI:** 10.3390/microorganisms13061279

**Published:** 2025-05-30

**Authors:** Zhichao Zhang, Zaozhu Niu, Zhan Chen, Yanzhuo Zhao, Lili Yang

**Affiliations:** Shijiazhuang Institute of Pomology, Hebei Academy of Agricultural and Forestry Sciences, Shijiazhuang 050061, China; zhangzc927@163.com (Z.Z.); niuzaozhu0311@163.com (Z.N.); chenzhan_elle@126.com (Z.C.); zyz432@126.com (Y.Z.)

**Keywords:** *Plasmopara viticola*, infestation cycle, effectors, integrated control

## Abstract

Downy mildew is among the most destructive diseases affecting grape production worldwide. It severely restricts the advancement of the grape industry. The causative pathogen, *Plasmopara viticola*, is an obligate biotrophic oomycete. Since the disease was introduced to Europe via grape cuttings in the 1870s, downy mildew has spread globally, resulting in devastating economic consequences. We review the current knowledge on the causative agent of grape downy mildew, its pathogenic mechanism, and control measures. Finally, we provide recommendations for developing more cost-effective strategies involving resistance genes and biocontrol agents to control grape downy mildew.

## 1. Introduction

Grapevine (*Vitis vinifera* ssp. *vinifera*) (family: Vitaceae) comprises approximately 60 inter-fertile wild *Vitis* species distributed in Asia, North America, and Europe under subtropical, Mediterranean, and continental-temperate climatic conditions [1]. Its cultivation and domestication appear to have occurred between the seventh and the fourth millennia C.E. in an area between the Black Sea and Iran [2,3,4,5]. Three main clades have been identified within subgenus *Vitis*, and data suggest that one clade is distributed in North America, another in Europe, and a third in East Asia [6]. The fruit contains various nutritive components, including amino acids, lipids, vitamins, and flavonoids [7,8,9,10,11,12]. In addition to being eaten fresh, grapes serve as crucial raw materials for raisins, fermented beverages, canned food, vinegar, and juice. During their growth period, grapes are exposed to various adverse factors such as pathogens and insects, which can result in various diseases or the consumption of the plants by pests. One pathogen, grape downy mildew, causes a serious disease affecting grape production globally.

Grape downy mildew was first reported in the northeastern United States. De Bary described the sexual and asexual stages of the causative pathogen, placed it in the genus *Peronospora*, and named it *Peronospora viticola* [13]. In 1888, Berlese and deTonir renamed this fungus-like eukaryotic microorganism *Plasmopara viticola* according to Schröder’s classification system [14,15]. In 1870s, probably after American grape cuttings had been introduced to Europe to replant vineyards destroyed by phylloxera, grape downy mildew emerged as a major problem for European viticulture [16]. Compared with wild American grapevines, which have strong immunity to downy mildew, European grape varieties are highly susceptible to *P. viticola* due to a lack of co-evolutionary history with the pathogen [17,18]. Successful invasive populations in Europe then served as the source for secondary introductions into other grape-growing regions such as Northeast China, South Africa, and Australia [16,19]. Reportedly, this disease affected 70% of French grape production in 1915, and it has been documented or reported in 96 countries and found on every continent except Antarctica [16].

*P. viticola* belongs to the Oomycota, Oomycetes, Peronosporales, and *Plasmopara*. It infects grape leaves and tender tissues, including shoots, inflorescences, spike-stalk, petioles, and berries. Overwintered oospores develop into microsporangia, from which the asexual zoospores are released [20]. Zoospores reach grapevine tissues via rain splash, air turbulence, and dew accumulation and produce germ tubes that invade the host through stomata and colonize the leaf parenchyma intercellularly (Figure 1). The mycelium of the pathogen absorbs nutrients from host cells using intracellular haustoria [16,21]. The infected leaves often develop yellow, water-soaked lesions in the initial infection stage. Subsequently, a white mould layer forms on the backs of the leaves (Figure 2). When the disease is severe, the lesions expand and merge, forming one larger, irregular lesion. With the expansion of lesions, the reduction in photosynthetic rates negatively affects sugar accumulation in berries and overwintering buds and delays berry ripening, which results in poor fruit set and quality and eventually leads to immense production loss [22]. The development of disease is favoured by mild temperatures and high humidity, although the pathogen seems to be becoming more tolerant to hot and dry weather [23,24]. In China, downy mildew remains the most devastating disease affecting grape production and is especially severe in vineyards under open cultivation. For instance, the majority of asymptomatic grape leaves (60%), leaf residues (80%), and soil samples (100%) tested positive for *P. viticola* in Ningxia Hui Autonomous Region, northwestern China [25]. Additionally, studies have revealed that the *P. viticola* strains collected from Chinese grape-growing regions exhibited substantial differences in terms of pathogenicity and long-distance dispersal of oospores [26]. This article reviews the current knowledge on the pathogenic mechanism of *P. viticola* and strategies for disease control, aiming to offer new ideas for controlling grape downy mildew and ensuring sustainable agricultural practices.

## 2. Life Cycle of *P. viticola*

*P. viticola* produces both sexual and asexual spores, which are responsible for the primary and secondary infection cycles, respectively [27]. In grape downy mildew, initially, a restricted number of germinating oospores cause infections early in the grapevine growing season. Then, pathogen produces sporangia containing asexually zoospores [28,29]. The pathogen overwinters by producing oospores on the fallen leaves, on berries, and in the soil, which indicates that oospores are its overwintering structures [27,29,30]. Oospores are rounded in shape and yellow, with a thick wall. They are always concentrated in the centre of the lesions [31]. Oospores represent the pathogen’s sexual stage because they are generated through the fusion of an oogonium and antheridium in host leaves [32]. Both environmental and endogenous factors affect oospore germination [30]. Among the environmental factors, rain plays a crucial role by providing moisture for oospore germination and moisturizing the soil surface to facilitate zoospore ejection, thereby promoting infection. Rain splashes can also disperse zoospores from the ground to the plant [27,33]. Sexual spores can be produced at any temperature, but they seem to be preferentially produced under dry conditions. Only mature oospores germinate rapidly, forming sporangia once exposed to a water film [16]. Rossi et al. studied how the germination dynamics of *P. viticola* oospores are affected by environmental conditions and established a consistent model of the general relationships between the germination dynamics of an oospore population and weather conditions [34]. Suitable humidity conditions combined with temperatures of 10 °C or above promote the development of oospores into microsporangia, from which biflagellate motile asexual zoospores are released [16,35]. Zoospores swim toward the stomata and produce germ tubes with which they penetrate leaves through the stomata to extract nutrients from grapevines [16,21]. The release of zoospores, as well as infection, occurred over a wide range of temperatures, but peak activity was observed at 15–20 °C. High nighttime temperatures also promote zoospore germination by attenuating the constitutive plant defence response and enhancing early infection by *P. viticola* [36,37] (Figure 3).

## 3. Pathogenic Factors of *P. viticola*

Effectors are a pathogen-secreted class of proteins or small molecules that facilitate infection and/or trigger defence responses by altering the cell structures or metabolic pathways of host plants [39,40,41]. Plant pathogenic oomycetes secrete several effector molecules to boost pathogenicity, and these molecules are primarily classified as apoplastic and intracellular effectors [40,42,43]. According to previous studies, RXLR and CRNs are two key effector types in pathogenic oomycetes [44]. These two types of effectors contain modular structures, including an N-terminal signal peptide that transports effector proteins into cells and some conserved motifs [42,45,46,47,48,49].

RXLR effectors are part of the largest group of proteins translocated from oomycetes. These effector proteins of pathogens are characterized by conserved N-terminal amino acid sequences, and they carry a similar highly conserved motif: the Arg-Xaa-Leu-Arg (RXLR) motif [50,51]. The release of genome sequences of *P. viticola* revealed that RXLR proteins accounted for a large proportion of the proteins in the secretomes, with a total of 540 RXLRs, accounting for about 33.9% of the total putatively secreted proteins [52]. Mestre et al. successfully produced expressed sequence tags (ESTs) from *P. viticola* by creating a cDNA library from germinated zoospores, identifying a total of 54 ESTs of putative secreted hydrolytic enzymes and effectors [53]. Later, following parallel transcriptome sequencing of *P. viticola* and *P. halstedii*, a *Plasmopara* species cDNA database (PlasmoparaSp) containing 46,000 clusters was released. Sequence analyses revealed the presence of 55 RXLR families [54]. Additionally, a 101.3 Mb whole-genome sequence of *P. viticola* has been reported [55]. In total, 1301 putative secreted proteins, including 100 putative RXLR effectors and 90 CRN effectors, have been identified through genome analyses [55]. The RXLR domain of the RXLR family effectors mediates the transport of effector molecules to plant cells. However, other studies have reported that the RXLR sequence of native AVR3a is cleaved before the protein is secreted into infected potato plants by the pathogen *Phytophthora infestans*. This indicates that the RXLR motif aids in AVR3a secretion by the pathogen, rather than playing a direct role in its entry into the host cell [48,50,56]. In addition, it has been found that PvRXLR131 interacts with host *V. vinifera* BRI1 kinase inhibitor 1 (VvBKI1) and its close homologs in *Nicotiana benthamiana* (NbBKI1) and *Arabidopsis* (AtBKI1) [57]. Chen et al. analysed 26 effectors from *P. viticola*, one of which, PvAVH53, interacted with the grapevine nuclear import factor importin alphas (importin-αs) (VvImpα and VvImpα4). Further studies have demonstrated that importin-αs are required for nuclear localization and that the function of PvAVH53 is essential for host immunity [58,59]. Additionally, another study identified PvRXLR111, a cell-death-inducing effector. On interaction with the *V. vinifera* putative WRKY transcription factor 40 (VvWRKY40), the stability of PvRXLR111 increased. This suggests that VvWRKY40 negatively regulates plant immunity and that PvRXLR111 suppresses host immunity by stabilizing VvWRKY40 [60,61]. A nucleus-localized effector, PvAvh74, has been identified as inducing cell death in *N. benthamiana* leaves. The two putative N-glycosylation sites and 427 amino acids of the PvAvh74 carboxyl terminus are necessary for the cell-death-inducing activity of PvAvh74 [62]. The effector protein RxLR50253 was found to attenuate plant immunity by decreasing H_2_O_2_ accumulation through its interaction with VpBPA1 in the plasma membrane [63]. Moreover, PvRxLR28 significantly augmented the susceptibilities of grapevine and tobacco to pathogens [64]. Transcription of the effector RXLR31154, induced in the early stage of infection, could promote leaf colonization by *P. viticola* and *Phytophthora capsici* in *V. vinifera* and *N. benthamiana*, respectively [65]. PvRxLR16, a nuclear-localized effector, directly triggered cell death in *N. benthamiana*. When transiently expressed, PvRxLR16 in *N. benthamiana* augmented the expression of defence-associated genes and promoted the accumulation of reactive oxygen species (ROS) to induce resistance to *P. capsici* in plants as well [66].

Crinkler or crinkling- and necrosis-inducing protein (CRN) effector proteins are a class of modular proteins that are translocated into the host cells. These effectors were first identified in *P. infestans* and are able to suppress plant defences [67]. The conserved N terminus of CRN harbours a distinct LXLFLAK motif, which is followed by a conserved DWL domain [68]. In *P. viticola*, the effector PvCRN17, which exerts a virulent effect on *N. benthamiana* and grapevine plants, was further investigated. Protein–protein interaction experiments revealed the grapevine VAE7L1 as a target of PvCRN17 [69,70]. Furthermore, when stably expressed, PvCRN11, an effector, inhibited *P. viticola* colonization in grapevines and *P. capsici* colonization in *Arabidopsis*, indicating that PvCRN11 may serve as a protectant against the disease and thus could be used to increase grape production [18].

Although the functions of several effectors have been reported, the molecular mechanisms underlying infection by the pathogen remain to be thoroughly elucidated (Table 1).

## 4. Prevention of Grape Downy Mildew

### 4.1. Physical Control

Physical control is an environmentally friendly method that aligns with the requirements of organic fruit production. Preventive measures such as mulching under grapevines can reduce the movement of primary zoospores from the soil to host plants [71]. Other physical control strategies for this disease typically involve scraping to remove diseased tissues and decreasing moisture content in vineyards by methods such as rain-shelter cultivation. Fertilization also affects *P. viticola* infection. Research has found that in plots treated with 0.5 and 0.6% foliar fertilizers and 200 kg phosphorus and potassium active substance per hectare, the rate of attack by the disease decreased to 0.01% of that seen in the untreated control plots (0.13%) [72]. Besides the methods mentioned above, the vertical shoot-positioning training system is also used to reduce the severity of downy mildew [73]. According to Rumbolz et al. [74], irradiation with white light, near-UV light (310–400 nm), or green light (500–560 nm) at intensities of >3–3.5 W/m^2^ can inhibit the production of sporangia [75]. Further studies have found that UV-C irradiation significantly reduces the area of *P. viticola* infection-induced lesions under greenhouse conditions [76].

### 4.2. Chemical Control

In 1882, Millardet discovered the protective effect of copper against grape downy mildew, observing that the copper-treated plants showed no symptoms while other parts of the vineyard developed symptoms. He recommended this treatment to protect grapevines against the disease [77]. Since then, several chemical formulations have been found to control downy mildew. Phenylamide (PA) fungicides, including metalaxyl, benalaxyl, and furalaxyl, are a widely used family of fungicides. Among them, metalaxyl is a representative PA that inhibits RNA biosynthesis [78,79]. As a result of the development of the fungicide industry, carboxylic acid amide (CAA) fungicides were developed in the 1980s that have no cross-resistance to phenylamide fungicides [80,81]. Dimethomorph is a systemic oomycete fungicide and is the first successfully developed CAA-type fungicide [81]. The fungicide strongly inhibits zoospore encystment, cystospore germination, and mycelial growth in vitro, rather than affecting zoospore discharge from sporangia [82]. Another CAA-type fungicide, benthiavalicarb, exerts the same effects mentioned above [83,84]. Although CAA fungicides have been widely used, the biochemical mode of action is still unclear [85]. Studies revealed that the potential targets include phospholipid biosynthesis [86] and cell-wall deposition [87,88]. In the late 1990s, Qo-inhibitor (QoI) fungicides such as azoxystrobin, famoxadone, and fenamidone were developed [85,89]. These fungicides inhibit mitochondrial respiration by binding to the Qo site of cytochrome b in the mitochondrial bc1 enzyme complex, thereby exerting high efficacy in controlling infections in the initial developmental stages and decreasing the inoculum potential [90]. When they were applied 1 to 5 days before inoculation, these fungicides provided 100% disease control at a concentration of 250 µg/mL [91]. On measuring the sensitivity of oospores differentiated in vineyards treated and untreated with azoxystrobin, Toffolatti et al. noted that fewer QoI-resistant strains could be recovered from a vineyard that had been treated with a azoxystrobin and folpet mixture than from a vineyard that had been treated with the QoI fungicide alone [92]. Those fungicides mentioned above are mainly single-site fungicides. In addition, fungicides with multisite activity, such as mancozeb (as a protectant), chlorothalonil, and copper compounds, have been intensively used to manage the disease [93,94,95] (Table 2). Mancozeb provided complete control of the disease when applied before inoculation with *P. viticola* and also exhibited moderate-to-high antisporulant activity when applied in postinfection and postsymptom modes, but this fungicide showed little ability to reduce disease incidence postinfection [91]. Chlorothalonil is another specific, non-systemic fungicide that, if applied during the first few days postexposure, may prevent foliar infections [96]. While Bordeaux mixture is a long-standing fungicide, many other copper compounds have been developed in recent decades. Pharmaceutical corporations have been fabricating copper-based fungicides in soluble forms as sulphates, oxychlorides, acetates, carbonates, oleates, silicates, and ohydroxides. They are widely used in pre-infection applications to form a protective barrier at the plant surface [97]. However, the wide application of inhibitors has resulted in a loss of fungicide sensitivity in *P. viticola* populations in some viticultural regions. Sensitive pathogen populations gained resistance via the selection of resistant mutants on exposure to fungicide applications. Since the finding of a high risk of resistance to PA and CAA fungicides in *P. viticola*, other studies have also reported widespread QoI resistance in *P. viticola* [81,93,98,99,100,101]. Furthermore, the excessive use of copper compounds results in toxicity to grapevines [102]. Thus, safer and more effective methods for control of the disease must be identified.

### 4.3. Biological Control

Copper-based control agents are generally non-degradable and thus accumulate in soil and water, altering water quality, spreading throughout the food chain, and eventually leading to tangible consequences related to human health [103,104]. Moreover, several studies have reported pathogen resistance to chemical fungicides. Hence, biological control has been regarded as an alternative means to replace chemical agents [88,105,106]. In general, beneficial microorganisms interact with plant pathogens through direct or indirect mechanisms. Consequently, several strains have been selected to control oomycete-induced diseases [107,108]. *Trichoderma* spp. are a type of filamentous fungi that colonize the rhizosphere and phyllosphere, thereby promoting plant growth and antagonizing numerous foliar and root pathogens [109,110]. Treatment of susceptible grapevine cultivars with *Trichoderma harzianum* T39 has been reported to induce grapevine resistance to downy mildew without negative effects on plant growth, and the strain has been developed into commercial products to combat *P. viticola* in practice [111,112,113]. As a type of arbuscular mycorrhizal fungus, *Rhizophagus irregularis* suppresses the expression of the *P. viticola* effector PvRxLR28 [114]. Furthermore, the bacterium species *Lysobacter capsici* can resist copper by producing copper oxidase and copper-exporting PIB-type ATPase. Thus, application of this microorganism in combination with low doses of a copper-based fungicide can increase the efficacy of control of grapevine downy mildew [115]. *Bacillus* is a widely studied biocontrol agent, and *B. subtilis* strain KS1, which was isolated from grape berry skin, can inhibit the pathogen [116]. *Alternaria alternata* (Fr.) Keissl. is an endophytic fungus isolated from grapevine leaves and is efficacious in controlling downy mildew by production of toxic diketopiperazine metabolites [117]. Furthermore, two bacterial strains, GLB191 and GLB197, identified as *B. subtilis* and *B. pumilus*, respectively, exhibited a robust protective effect against *P. viticola* in the field [118]. Strains belonging to *Curtobacterium herbarum*, *Thecaphora amaranthi*, and *Acremonium sclerotigenum* also exhibit high biocontrol efficiency against *P. viticola* [119,120] (Table 3). Despite having great development value, there are limitations of biological control strains, such as the need to ensure their persistence following deployment. Environmental factors including soil types, precipitation, temperature, humidity, host microorganisms, and plant leaf texture affect the colonization of strains [121,122]. Thus, discovering new biocontrol control strains, field-testing effective strains, and developing combinations of various strains are problems worthy of attention in further research [121].

In addition to beneficial microorganisms, plant-derived active substances are often used in disease control because they are environmentally friendly and safe to animal and human systems [123]. Stilbenoids from grape cane are plant-defence-related bioactive compounds. The stilbenes content varies widely and is influenced by genetic and environment factors [124,125]. Two stilbenoid compounds, δ-viniferin and pterostilbene, are toxic to the pathogen’s zoospores, affecting their mobility and thus inhibiting disease development [126]. Moreover, β-1,3-glucan laminarin, derived from the brown algae *Laminaria digitata*, is an elicitor of defence responses in grapevine. It effectively reduces *P. viticola* growth and development on infected grapevine plants. This compound inhibits 75% of the infection caused by the pathogen in grapevine plants [127]. Moreover, several compounds belonging to capsaicinoids and polyphenols have been identified from chili pepper (*Capsicum chinense* Jacq.) pod extract. These compounds from this extract significantly inhibited *P. viticola* growth and sporulation [128]. Using gas chromatograph-tandem mass spectrometry, four compounds, namely 4-allylpyrocatechol, eugenol, alpha-pinene, and beta-pinene, were identified from the methanol extract of *Piper betle* leaves. Treatment with 4-allylpyrocatechol combined with eugenol, alpha-pinene, or beta-pinene augmented the inhibitory effect on grape downy mildew and completely suppressed the disease [129]. Furthermore, two compounds including pinosylvins and pinocembrin that were identified from pine-knot extract significantly inhibited zoospore mobility and mildew development. These findings strongly suggest that pine knot can be applied as a natural antifungal product [130] (Table 4).

Applying various agents to plants can induce resistance to subsequent pathogen invasion, both locally and systemically [131]. The induced resistance allows broad-spectrum disease control and involves the resistance mechanisms of plants. Hence, researchers are exhibiting great interest in developing agents that mimic natural inducers of resistance [132]. Protein hydrolysates from soybean and casein trigger resistance in grapevine plants against *P. viticola*. The application of soybean and casein reduced the area of the infected leaf surface by 76% and 63%, respectively, on *V. vinifera* cv. Marselan plants compared with the control [133].

### 4.4. Breeding Disease-Resistant Grape Varieties

Breeding for resistant cultivars is a method of selective breeding used to reduce the effects of biotic stresses in crops. The aim is to develop offspring that inherit the increased tolerance of one parent and the desirable crop qualities of another [20]. Screening and cultivating disease-resistant varieties has been considered the most efficient and feasible method for mitigating grape downy mildew-induced damage. Wild grapevine species are reliable sources of resistance to many diseases and environmental stresses affecting cultivated grapevines [134]. Compared with susceptible cultivars, resistant cultivars always exhibit a hypersensitive reaction, with a faster reduction in the photosynthetic rate and increased production of H_2_O_2_ observed within hours of inoculation [135,136,137]. European *V. vinifera* cultivars were highly susceptible to *P. viticola*, whereas *Muscadinia* species and several American and Asian *Vitis* species displayed varying levels of resistance against the pathogen. Attempts to produce grapevine cultivars carrying resistance genes against downy mildew have rveraled that the resistance trait is quantitatively inherited [134]. Quantitative trait loci for resistance against downy mildew have been identified [134,138,139,140,141]. Molecular maps based on populations segregated on the basis of disease-resistant traits, including resistance to downy mildew, are available [141,142,143]. Deployment of resistance genes is an effective, environmentally favourable, and commonly used strategy for conferring disease resistance to crop plants. Several resistance genes against downy mildew have been cloned, such as the *RPP* series from *Arabidopsis* [144] and *Dm3* from *Lactuca sativa* [145]. Most of the characterized plant R genes encode proteins with leucine-rich repeat domains, a central nucleotide binding site, and a variable N terminus composed of a Toll/Interleukin-1 receptor domain or a coiled-coil domain [146]. The wild North American grapevine species *M. rotundifolia* is resistant to both powdery and downy mildew. Accordingly, a typical TIR-NBS-LRR-type disease-resistance gene, named *MrRPV1*, has been cloned from *M. rotundifolia*. Anderson et al. screened a bacterial artificial chromosome (BAC) library constructed from the genomic DNA of a single progeny plant (BC5:3294-R23). This progeny plant contained the MrRPV1 locus, the resistance locus located in a region on chromosome 12, identified using a marker derived from BAC-end sequences [147]. Additionally, *MrRPV1* conferred resistance to multiple downy mildew isolates from France, North America, and Australia. Comparisons of gene organization and coding sequences between *M. rotundifolia* and the cultivated grapevine *V. vinifera* at the *MrRPV1* locus unveiled a high synteny level, suggesting that TIR-NB-LRR genes at this locus share a common ancestry [148]. Besides, *Rpv3* has been identified as a single major resistance locus for most of the downy mildew-resistant cultivars found in Europe. *Rpv3*-mediated resistance is associated with a defence mechanism that triggers the synthesis of stilbenes and programmed cell death (PCD), resulting in reduced, but not suppressed, pathogen growth and development. A dense cluster of NB-LRR genes at the *Rpv3* locus provides a distinctive advantage that allows native North American grapevines to withstand downy mildew [149,150]. Further research suggested that *Rpv12*^+^ has an additive effect with *Rpv3*^+^ that protects vines against natural infections and confers foliar resistance to strains that are virulent on *Rpv3*^+^ plants [151]. Additionally, *Rpv10* was initially introgressed from *V. amurensis*, a wild species of the Asian *Vitis* gene pool, and it provides an addition source of genes from which grape breeders can develop new resistant cultivars [152]. VDAC3 from *V. piasezkii* ‘Liuba-8’ is considered to interact with VpPR10.1. Coexpression of VpPR10.1/VpVDAC3 induced cell death in *N. benthamiana*. This suggests that the VpPR10.1/VpVDAC3 complex triggers a cell-death-mediated defence response to *P. viticola* in grapevines [153]. VvNAC72 is a transcription factor containing the NAC (NAM/ATAF/CUC) domain and is located in the nucleus. The pathogen induces *VvNAC72* expression, which directly binds to the *VvGLYI-4* promoter region via the CACGTG element, thereby inhibiting *VvGLYI-4* transcription and leading to increased methylglyoxal content. This increases ROS production and confers stronger resistance to pathogen invasion [154]. Long non-coding RNAs (lncRNAs) are regulatory transcripts of length > 200 nt. These lncRNAs are considered crucial nodes in plant antifungal defence networks. Based on RNA-seq data, 83 downy mildew-responsive lncRNAs were identified, and 17 lncRNAs were found to be co-expressed with eight transcription-factor families, including those related to stress responses, such as C_2_H_2_, ERF, HSF, GRAS, C_3_H, and NAC. The identified lncRNAs can be further examined and leveraged as candidates for biotechnological improvement or genetic breeding to improve resistance to fungal disease in grapevines [155].

## 5. Conclusions and Future Directions

Grape downy mildew is a destructive disease caused by the oomycete pathogen *P. viticola*. This pathogenic microorganism can adapt to and overcome the immune system of grapevines, and its ability to release a range of molecular weapons, such as effector proteins, during plant infections significantly contributes to their success. Although many solutions have been proposed, a straightforward tool to optimize disease-management protocols is still lacking. This review summarizes the pathogenic mechanisms of grape downy mildew, a disease caused by *P. viticola*. In addition, the review discusses various approaches used to control the disease. For disease control, it is imperative to develop an accurate monitoring and early-warning technology. Since the physiological metabolisms and ultrastructures of the leaves differ between healthy and infected plants, it is feasible to monitor the occurrence of grape downy mildew and conduct dynamic analyses to provide early warning of infection based on physiological and biochemical indicators. These techniques can also provide technical support to develop scientific approaches to the prevention of downy mildew in grape.

Breeding resistant grapevine cultivars is the most sustainable tool for crop protection. Research into breeding resistant cultivars should continue to support the timely introduction of new resistant cultivars to the market. This article highlights the urgent need for comprehensive studies to identify resistance genes, not only by using genetic markers associated with disease-resistant germplasms, but more importantly, by analysing the aspects of the host immune system that are regulated by *P. viticola* pathogenic factors. Newer approaches such as immunity-inducing agents and broad-spectrum biocontrol agents must be investigated and applied when feasible. These biological agents, which have few or no ecological and environmental effects, can be useful in sustainable agriculture and effectively meet the requirements of safe, high-quality, and pesticide-free food production. There is no approach that can permanently eliminate grape downy mildew, and integrated disease-management programs that optimize all available treatment options will help reduce the negative impacts of the disease on growers.

## Figures and Tables

**Figure 1 microorganisms-13-01279-f001:**
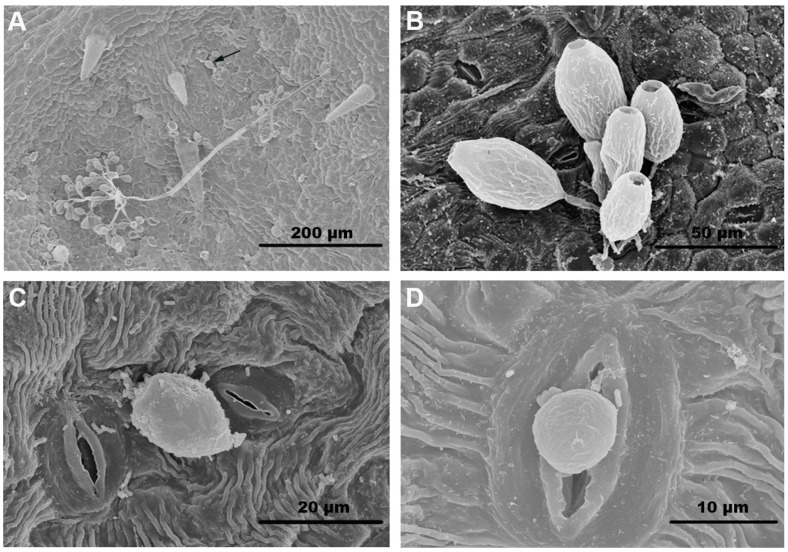
Scanning electron micrographs of grapevine leaves infected by *P. viticola*. (**A**) Sporangiophores and sporangia of *P. viticola*; the position indicated by the arrow indicates a zoospore with a germ tube. (**B**,**C**) Sporangia. (**D**) An encysted zoospore infecting a leaf through a stoma.

**Figure 2 microorganisms-13-01279-f002:**
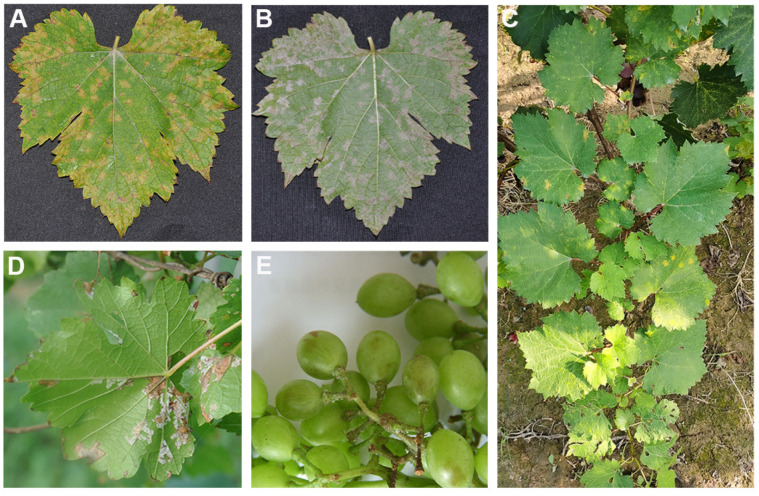
Symptoms caused by *P. viticola*. (**A**) Lesions on the surface of a grapevine leaf. (**B**) Pathogen development on the bottom of a grapevine leaf. (**C**,**D**) Infected grapevines in a vineyard. (**E**) Peduncle of grapes infected by *P. viticola*.

**Figure 3 microorganisms-13-01279-f003:**
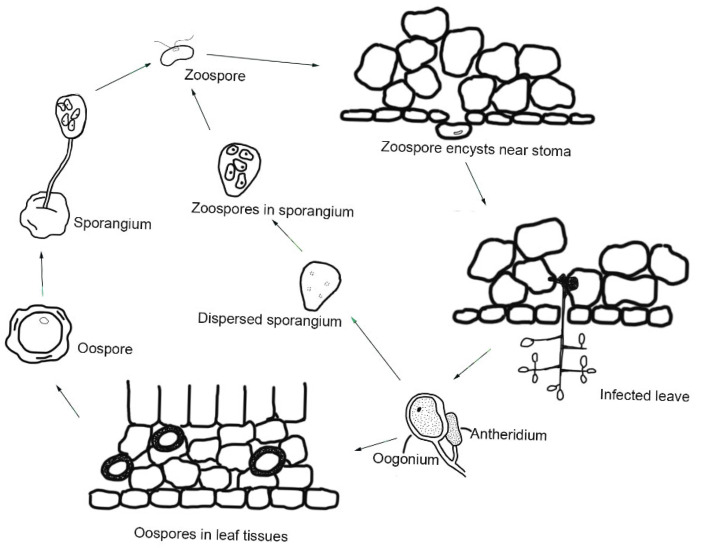
The life cycle of *P. viticola*. This figure generated based on the reference of [38] with modifications.

**Table 1 microorganisms-13-01279-t001:** Partial list of pathogenic secretory proteins of *P. viticola* involved in disease processes.

Effector	Virulence Subprocesses	Subcellular Locations	Plant Interactors	Biochemical Activities ^a^	Cellular Processes Affected ^b^
PvRXLR131 (Lan et al., 2019 [57])	Suppresses defence-related callose deposition	Unknown	VvBKI1	Targets the BKI1 receptor inhibitor of growth- and defence-related BR and ER signalling	Suppresses BR and ER signalling in plants
PvRXLR111 (Liu et al., 2018 [60], Ma et al., 2020 [61])	Decreases H_2_O_2_ accumulation	Nucleus	VvWRKY40	Interacts with the negative immune regulator VvWRKY40	Suppresses PTI responses by targeting the VvWRKY40
PvAvh74 (Yin et al., 2019 [62])	Increases ROS accumulation	Nucleus	Unknown	Triggers cell death via SGT1, Hsp90, RAR1, NDR1, EDS1, and MAPK cascades	Activates SA, JA signalling pathways and ROS accumulation
PvRxLR28 (Xiang et al., 2016 [64])	Decreases H_2_O_2_ accumulation	Nucleus, cytoplasm	Unknown	Reduces the transcriptional levels of defence-related genes and impairs H_2_O_2_ accumulation	Represses SA, JA, and ET signalling pathways and H_2_O_2_ accumulation
PvRXLR31154 (Liu et al., 2021 [65])	Controls the ROS-mediated defence response and reduces defence response in planta and enhanced colonization	Chloroplast, cytoplasm, and nucleus	VpPsbP	Stabilizes PsbP during infection	Suppresses H_2_O_2_ production and activates ^1^O_2_-mediated signalling
PvRxLR16 (Xiang et al., 2017 [66])	ROS accumulation and immunity-associated pathways	Nucleus	Unknown	Triggers cell death depending on SGT1, Hsp90, and RAR1	Activates SA, JA, and ET signalling pathways and promotes ROS accumulation
PvAVH53 (Chen et al., 2020, 2021 [58,59])	Triggers cell death	Nucleus	VvImpα and VvImpα4	Targets VvImpα/α4	Interacts with VvImpα/α4 to regulate the immune response
RxLR50253 (Yin et al., 2022 [63])	Decreases H_2_O_2_ accumulation	Plasma membrane, Cytoplasm, and Nucleus	VpBPA1	Inhibits degradation of VpBPA1 protein	Decreases H_2_O_2_ accumulation by targeting and stabilizing VpBPA1
PvCRN17 (Xiang et al., 2021, 2022 [69,70])	Suppresse defence responses	Mainly localized in the plasma membrane and nucleus	VvAE7L1 and VvNRPPII-X1	Competes with VCIA1 to bind with VAE7L1 and VAE7	Interrupts the maturation of Fe-S proteins and suppresses Fe-S proteins-mediated defence responses
PvCRN11 (Fu et al., 2024 [18])	Increases ROS accumulation	Nucleus, cytoplasm, plasma membrane	Unknown	Induces BAK1-dependent immunity in the apoplast, and PvCRN11 overexpression in intracellular induces BAK1-independent immunity	Increases ROS accumulation, promotes MAPK activation and PR1 and PR2 up-regulation

^a^ BR, brassinosteroid; ER, ERECTA; SGT1, suppressor of G-two allele of Skp1; Hsp90, heat shock protein 90; RAR1, Mla12 resistance; NDR1, non-race-specific disease resistance 1; EDS1, enhanced disease susceptibility 1; MAPK, mitogen-activated protein kinase. ^b^ PTI, PAMP-triggered immunity; SA, salicylate-mediated signalling pathway; JA, jasmonate-mediated signalling pathway, ET, ethylene signalling pathway; ROS, reactive oxygen species; PR, pathogenesis-related gene.

**Table 2 microorganisms-13-01279-t002:** Major inhibitors used to control *P. viticola*.

Fungicide Group	Common Name	Mode of Action	Reference
PA	metalaxyl	Inhibited biosynthesis of RNA	[78,79]
	benalaxyl		[78]
	furalaxyl		[79]
CAA	dimethomorph	Inhibited zoospore encystment, cystospore germination, and mycelial growth	[82]
	benthiavalicarb		[84]
QoI	azoxystrobin	Cytochrome bc1 Qo site	[85]
	famoxadone		[89]
	fenamidone		[85]
-	copper compounds	Multi-site fungicides	[85]
	mancozeb		[87]
	chlorothalonil		[85]
	folpet		[85]

**Table 3 microorganisms-13-01279-t003:** Partial list of biocontrol microorganisms used against grape downy mildew.

Biocontrol Strategy	Species Used	Possible Mode of Action	Reference
Fungi	*Trichoderma harzianum* T39	Modulated defence-related genes	[111,112,113]
	*Rhizophagus irregularis*	Modulated the expression of pathogenicity effectors	[114]
	*Alternaria alternata*	Antifungal metabolites diketopiperazines inhibited sporulation	[117]
	*Thecaphora amaranthi*	-	[119]
	*Acremonium sclerotigenum*	Inhibited the germination of sporangia	[119,120]
Bacteria	*Lysobacter capsici*	Produced biocontrol compound 2,5-diketopiperazine	[115]
	*Bacillus subtilis* KS1	Antifungal lipopeptide iturin A	[116]
	*Bacillus subtilis* GLB191	Antifungal metabolites	[118]
	*Bacillus pumilus* GLB197		[118]
	*Curtobacterium herbarum*	-	[119]

**Table 4 microorganisms-13-01279-t004:** Partial list of plant-derived active substances used against downy mildews.

Botanical Name	Plant Part	Substances Used	Possible Mode of Action	Reference
*Vitis vinifera*	cane, wood, and root	*r*-viniferin	Inhibited pathogen sporulation	[124]
		hopeaphenol		[124]
		*r*2-viniferin		[124]
		δ-viniferin	Affected zoospore mobility	[126]
		pterostilbene		[126]
*Laminaria digitata*	-	β-1,3-glucan laminarin	An elicitor of defence responses	[127]
*Capsicum chinense* Jacq.	pod	capsaicinoids and polyphenols	Inhibited pathogen’s growth and sporulation	[128]
*Piper betle*	leaves	4-allylpyrocatechol combined with eugenol, alpha-pinene, or beta-pinene	Inhibited *P. viticola* growth	[129]
*Pinus pinaster*	pine knots	pinosylvins	Inhibited zoospore mobility and mildew development	[130]
		pinocembrin		[130]

## Data Availability

No new data were created or analysed in this study. Data sharing is not applicable to this article.

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
