# Peer review of "Review of the Pathogenic Mechanism of Grape Downy Mildew (Plasmopara viticola) and Strategies for Its Control"

_microorganisms, 2025, doi:10.3390/microorganisms13061279_

Round 1
Reviewer 1 Report
Comments and Suggestions for Authors
Please find the attachment with minor suggestions and comments on improving the manuscript.
The manuscript "Review of the Pathogenic Mechanism and Control Strategies for Grape Downy Mildew" is comprehensive and well-designed. However, a few issues need to be addressed.
First, the manuscript title would be more informative for readers if after "...grape downy mildew" Latin name of an oomycete (Plasmopara viticola) will be placed in the parenthesis.
It would be good if the manuscript were checked and corrected by a native speaker because many simple mistakes can be found, and sentences can sometimes be hard to understand.
Tables with the control agents are necessary to facilitate the text in chapters about chemical and biological controls, and readers of this manuscript will appreciate them.
The Introduction and Conclusions sections are fine.

In my opinion, the manuscript should be read by a native speaker because some sentences are difficult to understand and there are grammar errors.
Reviewer 2 Report
Comments and Suggestions for Authors
The evaluated review discusses downy mildew of grapevine, from generalities of the causal agent, symptomatology in the plant, virulence factors and control strategies. The manuscript has coherence, however it is important to implement images and tables that improve the comprehension of the paper. The manuscript can be accepted, after minor revisions.
Comments:
1.-Please add a Figure showing the life cycle of the oomycete plasmopara viticola, remember that this is a review and images help to improve the understanding of the paper.
2.-Add a table focusing on the different treatments against grapevine downy mildew if the authors wish, they could focus only on biological control.
Reviewer 3 Report
Comments and Suggestions for Authors
This manuscript is devoted to the downy mildew of grapes, which is caused by Plasmopara viticola (Oomycota). It is an extremely serious disease of grape vines that can result in severe crop loss. This article reviews the current knowledge on the life cycle of causal agent, pathogenesis, and prevention of the diseases. Physical, chemical and biological control, and breeding for disease resistant grape cultivars are presented separately. More than 140 scientific articles have been used to present these problems. The manuscript has been prepared with care. Only in a few places could minor additions be made. In this review article some directions and ideas for obtaining better protection of grapevines against the disease are shown. The manuscript should be published in Microorganisms after taking into account the comments listed in Remarks.
Line 47 the taxonomy of Plasmopara viticola should be given
Line 74 it should be clarified what Figure 1D shows. Zoospores P. viticola are biflagellated. Here it is rather encysted zoospore form before entering the leaves through stomata (compare text in lines 101-102)
Line 85 and soil – requires clarification on soil or in soil?
Line 101-102 this text requires correction, there is a contradiction between the first and second part of this sentence
Line 177 is it possible to give examples of how fertilization affects P. viticola infection
Line 231 it should be better emphasized which of the fungicides mentioned in this section create the possibility of developing resistant strains of P. viticola. The negative aspects of using the listed fungicides should also be better highlighted
Line 252 Alternaria alternata – a specific strain should be given because many strains of Alternaria alternata show pathogenic properties for different plant species
Are there any cases of practical use of antagonistic fungi or bacteria in the fight against P. viticola
Line 315 Arabidopsis – it should be in italic
Line 381 integrated disease management program is only mentioned in the Conclusion. It should be considered whether it would be possible to devote a separate section to this problem, if there are any specific examples of combined actions.
Round 2
Reviewer 1 Report
Comments and Suggestions for Authors
The Authors have corrected the manuscript according to suggestions and comments.